

# Detecting rumors in social media using emotion based deep learning approach

Drishti Sharma[*] and Abhishek Srivastava[*]

Department of Computer Science and Engineering, Indian Institute of Technology Indore, Indore, Madhya Pradesh, India

[*] These authors contributed equally to this work.

## ABSTRACT

Social media, an undeniable facet of the modern era, has become a primary pathway for disseminating information. Unverified and potentially harmful rumors can have detrimental effects on both society and individuals. Owing to the plethora of content generated, it is essential to assess its alignment with factual accuracy and determine its veracity. Previous research has explored various approaches, including feature engineering and deep learning techniques, that leverage propagation theory to identify rumors. In our study, we place significant importance on examining the emotional and sentimental aspects of tweets using deep learning approaches to improve our ability to detect rumors. Leveraging the findings from the previous analysis, we propose a Sentiment and EMotion driven TransformEr Classifier method (SEMTEC). Unlike the existing studies, our method leverages the extraction of emotion and sentiment tags alongside the assimilation of the content-based information from the textual modality, *i.e.*, the main tweet. This meticulous semantic analysis allows us to measure the user's emotional state, leading to an impressive accuracy rate of 92% for rumor detection on the "PHEME" dataset. The validation is carried out on a novel dataset named "Twitter24". Furthermore, SEMTEC exceeds standard methods accuracy by around 2% on "Twitter24" dataset.

# INTRODUCTION

The sphere of social media is experiencing a significant surge. Social networking platforms are ubiquitous, seamlessly woven into our existence's fabric. Their application spans various sectors like marketing, user feedback, and business establishment, expanding remarkably. Virtually every aspect of our lives is influenced by them, whether it is a follow-up of a fashion trend or how we react to any information; everything now relates to how social media depicts it (*Heinrichs, Lim & Lim, 2011*). With a user population in the millions, as shown in Fig. 1, these platforms effortlessly bring people together with just a single click. Information propagation is expeditious *via* these platforms, which may lead to any side of the coin (*Camacho et al., 2020*).

The term "rumor" is frequently encountered in our everyday experiences. It refers to information whose accuracy has yet to be confirmed. Unsubstantiated details can

Corresponding author
Abhishek Srivastava,
asrivastava@iiti.ac.in

**Social Media Users on different platforms**

**Figure 1** **A schematic illustration of monthly active social media user around the world.** The unit of number on *y*-axis is 1 unit = 1 billion.

significantly affect individuals' well-being (*Allport & Postman, 1947*). For instance, there have been occurrences around the world, like in India, such as the mistreatment of Hindi-speaking laborers, preventing them from working in Tamil Nadu. Another instance involves the event of the 2022 Republic Day, where a peaceful procession turned into a violent outbreak and street riots in Delhi. Perpetrators exploit rumors to fulfill their aim of sowing disorder, fomenting riots, and causing destruction.

Rumors are closely intertwined with social media, utilizing its extensive user community as an ideal and rapid channel to disseminate unverified information to many individuals instantaneously. The COVID-19 period clearly illustrates this phenomenon, with false narratives spreading across platforms like Twitter (X), WhatsApp, and Instagram (*Dasgupta, Mishra & Yadav, 2021*).

Detecting social media rumors is crucial due to the potential consequences they can trigger, including loss of life, property damage, economic upheaval, and personal reputation harm. Various measures have been implemented to prevent the spread of rumors. A rumor's impact persists until its veracity is definitively established (*Gumaei et al., 2022*). Hence, effective detection of rumors is of utmost importance.

Existing studies have demonstrated that investigators have employed diverse machine learning techniques such as Naive Bayes and Random Forest, as well as deep learning methodologies like recurrent neural networks (RNN) (*Ma et al., 2016*), recursive neural networks (RvNN) (*Ma, Gao & Wong, 2018*), and graph convolutional networks (GCN) (*Bian et al., 2020*), to identify misinformation in data, collected from platforms such as

Weibo and Twitter. The deep learning methods show superior performance on tasks like classification and translation (*Pattanaik, Mandal & Tripathy, 2023*). In deep learning-based approaches, we investigated that the GACL (Graph Adversarial Contrastive Learning) method deals with loss function, AFT (Adversarial Feature Transformation) produces conflicting samples in order to detect rumor (*Sun et al., 2022*). The RNN-based method, employing three recurrent units, learns the hidden representations that encapsulate the variation in contextual information (*Ma et al., 2016*). Feature fusion models with a fusion layer were employed to detect rumor by utilizing only a few labeled instances (*Lu et al., 2021*). Further advancement led to bottom-up RvNN and top-down RvNN, which conform to the propagation layout of tweet (*Ma, Gao & Wong, 2018*). Contemporary methodologies place a stronger emphasis on comprehending the propagation patterns of rumors. In addition, methods such as Credible Early Detection (CED), which leverage timestamps, have been employed to timely identify rumors (*Song et al., 2019*).

One crucial aspect that may have received less attention than it deserves is the sentiment expressed within a tweet. The emotional undertones of a statement provide insights into the writer's state of mind. Tweets with high emotional value spread rapidly and are more likely to be perceived as rumors. A recent study shows the interconnectedness of fake news and sentiments (*Ajao, Bhowmik & Zargari, 2019*). As a result, we investigated the SAME (Sentiment-aware multimodal embedding) model that incorporates latent sentiment to detect fake news (*Cui, Wang & Lee, 2019*).

Our proposed research prioritizes the investigation of the emotional and sentiment dimensions of tweets shared online and its potential role in enhancing the accuracy of rumor prediction. To the best of our understanding, our Sentiment and EMotion driven TransformEr Classifier method (SEMTEC), assimilates the context-based information and leverages emotions and sentiments from the textual modality, is pioneering in its application to rumor detection by extensively considering semantic attributes. To establish the validity of our research, we conducted experiments using the publicly accessible ''PHEME'' dataset. Furthermore, we created a novel dataset named ''Twitter24'', which contains tweets from the social media platform ''Twitter (X)''. To ensure the accuracy of label assignment, a manual verification process is employed. The labels are verified *via* the fact-checking website Boom Fact Check. Our SEMTEC model demonstrated exceptional performance, yielding an accuracy of approximately 92% on the ''PHEME'' and exceeds standard methods accuracy by around 2% on the ''Twitter24'' dataset. Subsequent subsections summarize the research gaps in the literature and how the proposed SEMTEC is novel compared to them.

## Research gaps

Prior research indicates the usage of machine learning and deep learning models for rumor classification. The research gaps have been identified and are mentioned below:

- The emphasis of prior research was to deal with rumor detection as either a simple classification problem or extending it to utilize propagation for classification.
- The availability of relevant datasets was restricted to specific social media platforms and inaccessible to researchers globally.

- The semantic aspects of the tweet, including emotions, sentiment, and contextual understanding, should be given more importance.

The proposed work focuses more on analyzing the semantic aspect of the primary tweet. The SEMTEC (Sentiment and Emotion-driven Transformer Classifier) method utilizes textual features, sentiment, and emotion tags, extending the rumor detection task beyond a simple binary classification. The proposed work excels in the rumor detection task because semantic analysis plays a crucial role in identifying the hidden aspect of the tweet. Following the same, the novel framework utilizes the extraction of the emotion tags from the textual modality. The contributions of SEMTEC are listed in the subsequent section.

## Key contributions

The key contributions that make our SEMTEC model significant are summarised below:

- We present a novel dataset named "Twitter24", annotated with rumor and non-rumor labels. It consists of tweets extracted from a social media platform called Twitter, now X. We manually assign the labels after verifying them with a fact-checking website, *i.e.,* Boom Fact Check. This establishes the correctness of assigning labels to the tweets.
- This work introduces a novel emotion-based deep learning method named Sentiment and Emotion driven Transformer Classifier (SEMTEC) for rumor detection.
- The novel framework leverages the sentiment tags extracted from the available textual modality.
- This study incorporates an emotional aspect derived from a recurrent neural network (RNN)-based multilayer model, encompassing a diverse range of emotion classes.
- Extensive experimental analysis on the publicly accessible "PHEME" dataset and "Twitter24" dataset demonstrates that our proposed method, SEMTEC, addresses prior limitations and exhibits improved performance compared to existing models.
- We present a novel dataset named "EmoPHEME" annotated with emotion labels specifically designed to facilitate research in emotion extraction. This dataset offers researchers a valuable resource for training and evaluating their emotion extraction models. The original PHEME dataset solely focuses on rumor detection labels. This enriched dataset "EmoPHEME" is a byproduct of our work and opens up new avenues for research in emotion detection and analysis.

The paper is organized as follows: 'Related Work' examines the prior research. 'Problem Statement' establishes the problem statement. 'Proposed Methodology' outlines our proposed methodology, and 'Experimentation and Results' details the experimental setup and presents the results. 'Discussion' presents a discussion, and finally, 'Future work' outlines avenues for future exploration, followed by 'Conclusion' encapsulates the concluding remarks.

## RELATED WORK

Identifying rumors has consistently been a widely studied issue, with researchers striving to address it due to its direct impact on our society (*Zheng et al., 2021*). Several efforts

have been made in understanding and detecting rumors (*Pattanaik, Mandal & Tripathy, 2023*; *Song et al., 2019*). For the same, a number of methodologies have been employed, including those based on machine learning and deep learning. Considering the different approaches, we have divided this section into three subsections highlighting the work's key approach.

## Machine learning based approaches

This section exhibits the approaches based on machine learning for rumor detection. When rumor detection was introduced as a threat to society, it was considered a simple classification problem. *Bingol & Alatas (2019)* and *Joulin et al. (2016)* demonstrated using models like naive Bayes, logistic regression, Random Forest, and Hoeffding Tree. The work mainly focuses on identifying the best algorithm for classifying whether the tweet was a rumor. Further advancement led to classification using hot topic detection elaborated by *Yang et al. (2015)*.

## Deep learning based approaches

This section demonstrates the related approaches previously employed utilizing the deep learning models. The emergence of deep learning technologies has significantly impacted the research fields due to their impressive performance. The extensive research done by *Kumar & Carley (2019)* and *Bian et al. (2020)* prove that deep learning methods are more effective in classifying rumors. The researchers *Ma et al. (2016)* utilized recurrent neural networks (RNN) for learning hidden features to get contextual information with time and tree-structured Recursive Neural Networks to find similarities in structure, respectively. Furthermore, *Feng et al. (2023)* proposed a BiMGCL model utilizing the bi-directional graphs to structure the rumor events.

## Propagation based deep learning approaches

Propagation mode utilizes features related to the flow of rumor with deep learning techniques. As explained by *Bian et al., (2020)*, rumor propagation facilitates the identification of rumors. Furthermore, *Ma, Gao & Wong (2017)* demonstrated that propagation tree kernel structure was used to identify patterns between tree structures, whereas *Sun et al. (2022)* focused on extracting dissimilarity between features of transmitted information to detect rumor.

Table 1 summarizes the related work section, highlighting a few works along with the proposed method.

Prior research on rumor detection primarily framed it as a classification problem. Studies explored various machine learning and deep learning techniques to identify rumors within the data effectively. However, these approaches focused solely on the propagation of the information, neglecting the semantics of tweets. The proposed study emphasizes the semantic analysis of core tweets for rumor detection. The proposed SEMTEC method incorporates textual features, sentiment labels, and emotion tags, expanding rumor detection beyond simple binary classification. The datasets consist of English tweets extracted from a prominent social media platform, Twitter (X). The proposed method demonstrates superiority in rumor detection by leveraging semantic analysis to uncover

**Table 1  Table summarizing the previous works and proposed method.**

| Author | Method name | Description |
|---|---|---|
| *Joulin et al. (2016)* | FastText Classifier | Employs linear classifier following training. Focuses on utilizing machine learning methods for classification. |
| *Bingol & Alatas (2019)* | ML Classifier | Considers rumor detection as simple classification problem. Utilizes standard machine learning classifiers |
| *Ma, Gao & Wong (2019)* | GAN-GRU | Employs generator to introduce conflicting and uncertain perspective in original tweet. |
| *Bian et al. (2020)* | BiGCN | Incorporates propagation by up-down GCN and dispersion via bottom-up GCN for rumor detection. |
| *Lu & Li (2020)* | GCAN | Generates explanations highlighting the evidence from suspicious retweeters and the concerning words they use. |
| **This article** | **SEMTEC (Proposed)** | **Establishes relationship between the semantic properties of the tweet with its veracity** |

underlying tweet content. *Vosoughi, Roy & Aral (2018)* highlighted that fear, disgust, and surprise are often associated with false stories, while true stories evoke anticipation, sadness, joy, and trust. Aligned with these findings, the proposed framework incorporates emotion tag extraction from the textual modality.

# PROBLEM STATEMENT

Our problem can be formally described in aspects which are illustrated as follows. We frame our problem as a binary classification problem. Given a dataset $D$ comprising of $N_t$ tweet, represented by $T = \{T_i\}_{i=1}^{N_t}$, for each tweet $T_i$, $T_i^t$ represents the textual features, corresponding extracted emotion feature $E_i$ and sentiment feature as $S_i$, we need to predict $L_i$ such that $L_i \in L$, where $L \in \{0, 1\}$ denoting rumor or non-rumor and $E_i \in \{anger, fear, joy, love, sadness, surprise\}$ and $S_i \in \{positive, negative, neutral\}$.

# PROPOSED METHODOLOGY

The proposed methodology primarily focuses on analyzing the semantic characteristics of a tweet and demonstrates how the implied emotions and sentiments contribute to classifying the tweet into specific categories. As explained by *Ajao, Bhowmik & Zargari (2019)*, we observed that text conveying high emotional states, such as fear or anger, is more readily accepted as accurate. Following the same, the proposed SEMTEC method leverages the emotional and sentiment aspects for rumor detection. This section provides a comprehensive overview of the proposed model's architecture, its components, and the necessary steps to achieve the final label of the tweets.

The sections discussed later will discuss the component-wise explanation for the proposed flow as shown in Fig. 2.

**Peer**J Computer Science

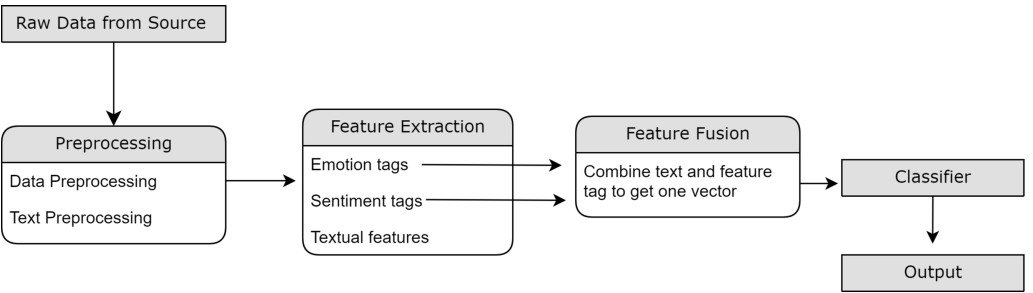

**Figure 2** **Illustration of flow of proposed methodology.**

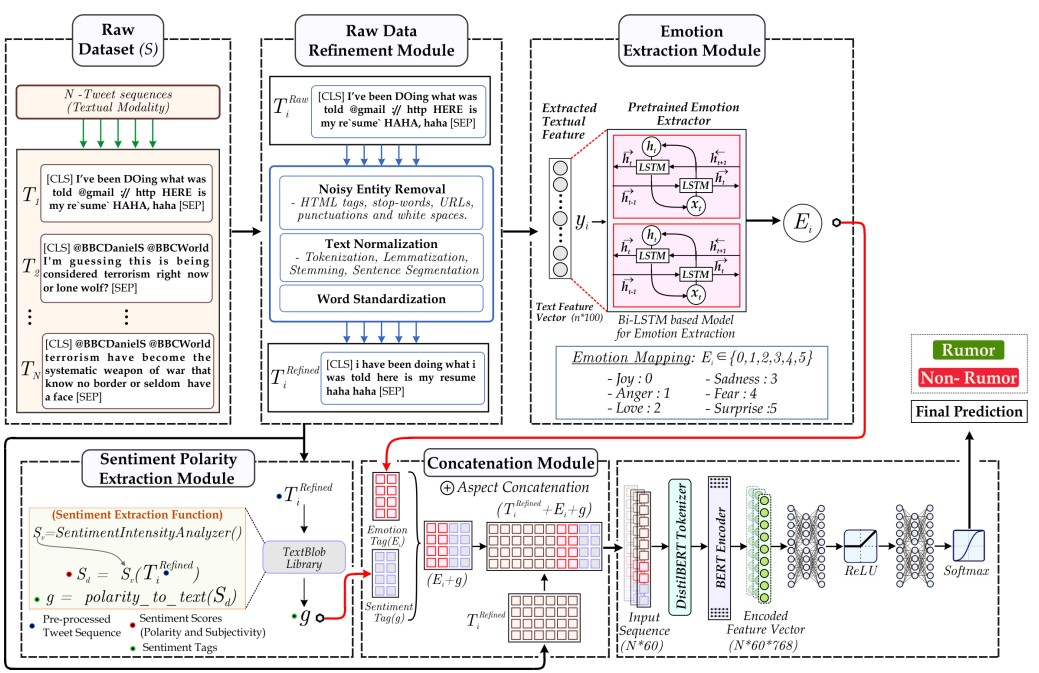

**Figure 3** **The image illustrates the proposed architecture.** The method follows the order of the Raw Dataset Module, containing tweets fetched from the source (Twitter), followed by the Data Refinement Module, which involves cleaning and pre-processing. Furthermore, feature extraction utilizing the Emotion Extraction Module and Sentiment Polarity Extraction module is done. Finally, the textual modality is combined with features and provided as input to the classifier to get a final label.

## Model architecture overview

This section outlines the proposed methodology. Figure 3 illustrates the overall architecture of our SEMTEC model.

- The proposed methodology utilizes the textual modality within the dataset. The textual modality must go through the pre-processing step to prepare the text for feature extraction. Textual pre-processing involves cleaning the text and pre-processing.
- Pre-processing is followed by a feature extraction module. 'Feature Extraction' discusses the textual feature extraction followed by sentiment and emotion modules.

- After acquiring the aforementioned features, we will examine their significant role in classifying a tweet as either a rumor or not. The emotion and sentiment tags are concatenated with the textual modality to create a comprehensive feature representation.
- Furthermore, the encoder transformer module is utilized to extract the contextual information from the tweet, followed by a classifier.

Subsequent sections provide a detailed decomposition of the proposed approach, outlining each component's function.

## Data refinement module

The task of rumor detection requires working with textual data that represents the way people typically talk. Human language is abundant with inconsistencies and errors that require rectification. The quality of NLP tasks depends significantly on the input data; the performance and accuracy of the model correspond to the accuracy of the input it receives. In this section, we will discuss the data refinement as mentioned in Fig. 3.

We have employed the text pre-processing toolkit ''text_hammer'' alongside a custom function to handle the text processing. The steps involved in data refinement are discussed below.

- Contraction of words: Contractions are abbreviations or shortened forms of usually two words involving an apostrophe. To provide a consistent meaning of a statement to the model, they are required to be expanded.
- Removal of emails, HTML tags, and special characters: Emails, HTML tags, and special characters increase the length of the text, which can hinder the extraction of necessary information from textual data.
- Handling accented characters: Since our model follows contextual-based learning, removing accented characters, *i.e.,* special symbols used to show a specific dialect or accent, will help us maintain a qualitative vocabulary corpus. Examples are résumé , naïve.
- Handling irregular capitalization: Proper capitalization facilitates the recognition of sentence tags such as nouns and pronouns, which leads to an easy flow in data mining.
- Lowercasing: To maintain the similarity and avoid additional vocabulary space for words with identical spelling, lowercasing is done. Example: Travel and travel have the same meaning, but when converted to vectors, both will have different values.

## Feature extraction

This section details the feature extraction modules depicted in Fig. 3. We employ various deep-learning techniques to extract features from the textual data.

### Textual feature extraction

The proposed work focuses on textual data, acknowledging its primacy in conveying meaning and context within social media posts. Therefore, to extract meaningful features from the textual content of tweets, we leverage the power of transformer-based deep learning models, explicitly employing the well-established BERT (Bidirectional Encoder Representations from Transformers) architecture (*Kenton & Toutanova, 2019*).

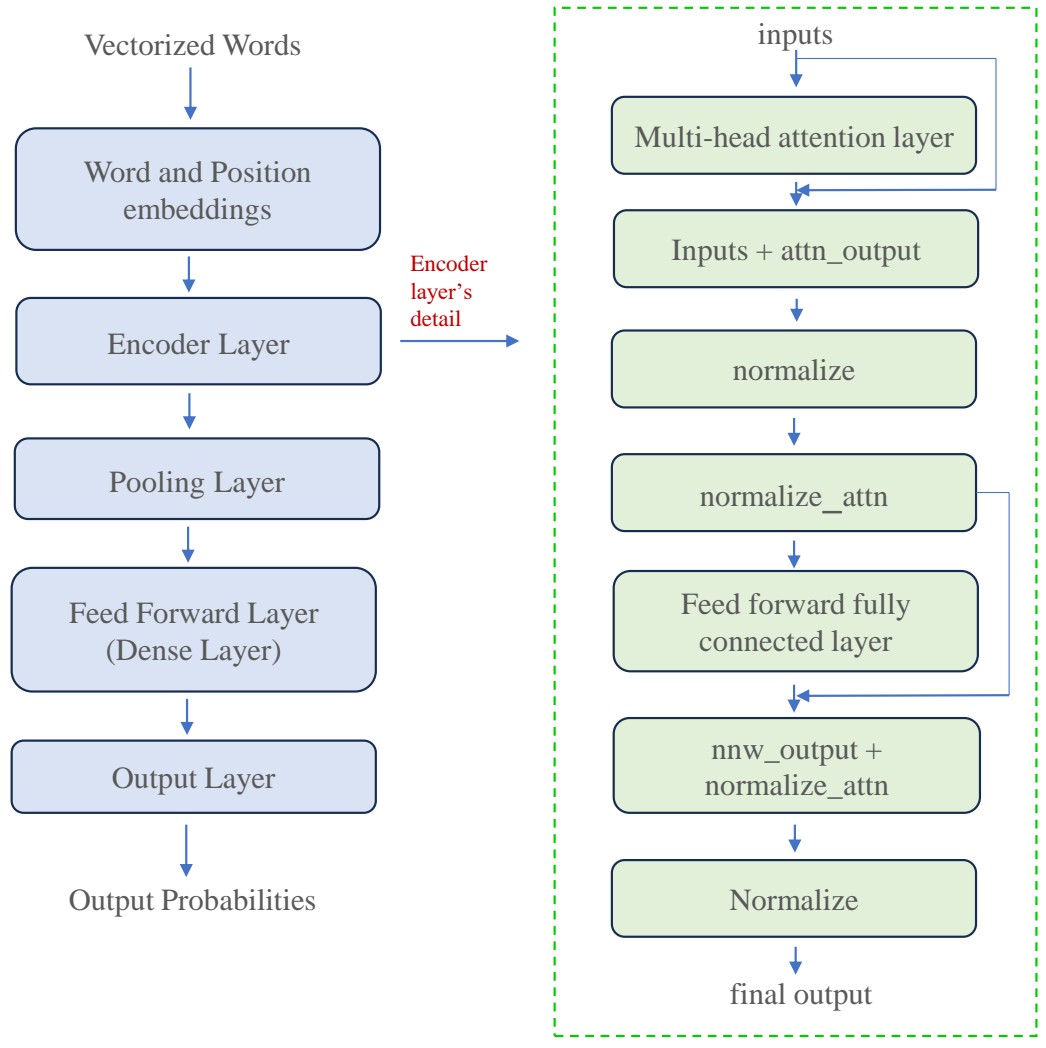

**Figure 4  Architecture of transformer-based deep learning model for embedding generation.**

The basic architecture of BERT is described in Fig. 4. The transformer architecture consists of encoder and decoder components. The encoder generates distinct continuous vector representations by processing the text of a tweet. The decoder then uses these vectorized embeddings to predict the desired outputs. The proposed work utilizes BERT because it is pre-trained on extensive data, around 3.3 billion words from Wikipedia and BooksCorpus. The model consists of a large number of encoder layers, a feed-forward network, and attention heads, as depicted in Fig. 4. The BERT language representation paradigm employs a deep architecture of stacked transformer encoder layers to generate contextualized word embeddings for each input token.

This study represents each tweet as a sequence of words denoted as $Wd_i = \{w_i^x\}_{x=1}^Z$, where $Z$ represents the word count in the tweet. These words sequence ($Wd_i$), forms the textual modality $(T_i^t)$ for a tweet $(T_i)$. We employ the distilBertTokenizerFast model from

the pre-trained transformer architecture to create the embedding representation. This tokenizer adds two special tokens, CLS (Class), at the beginning and SEP (Separator), at the end of each tweet's sequence.

For a given tweet $(T_i)$, we provide the input $(T_i^t)$. This input is then processed to generate a sequence of integer-based tokens $D_t$ shown in Eq. (1).

$$D_t = distilBertTokenizerFast(T_i^t) \tag{1}$$

As depicted in Eq. (1), the distilBERT tokenizer aids in getting tokens in the form of integer sequences denoted as $D_t$. For any tweet $(T_i)$, the output is tokens and can be demonstrated as $D_t = \left\{ d_i^x \right\}_{x=1}^{l}$, where $l$ denotes the length of sequence. In our work, we are taking a fixed sequence length of 60 for each tweet, *i.e.*, $l = 60$. Padding will be done for the tweets having a length of less than 60. Further, the tokens will be passed from the Encoder model to get the embedding vector for each token, as shown in Eq. (2).

$$E_m = BERT \left\{ d_i^x \right\}_{x=1}^{l} \tag{2}$$

Equation (2) illustrates the generation of the embedding vector $E_m$ when passed through the encoder model, *i.e.*, BERT. Here, $E_m = \{e_x\}_{x=1}^{d}$, where $d$ is the dimension of size 768. The demonstrated process was textual feature extraction.

### Sentiment feature extraction

The extraction of sentiment features from textual modality is substantiated to gather contextual insights into the tweet. To extract the sentiment tags, we implement a module from the natural language processing toolkit, namely TextBlob. The module is pre-trained on a variety of datasets.

TextBlob leverages a lexicon-based sentiment analysis approach and initially determines the intensity (positive or negative orientation) of individual words in a sentence. Lexicon-based approaches involve the use of a pre-built dictionary that categorizes words as positive or negative.

The generation of tags involves using a sentiment label analyzer as reflected in Eq. (3).

$$S_v = SentimentIntensityAnalyzer() \tag{3}$$

Furthermore, polarity scores are generated as shown in Eq. (4) followed by estimating tags from the calculated scores.

$$S_d = S_v.polarity\_scores(T_i) \tag{4}$$

Equation (4) represents the use of the analyzed score in the generation of the polarity score, which will be converted to provide us with the final sentiment label. The algorithm below outlines the procedure for sentiment tag extraction and subsequent incorporation into the original textual features.

### Emotion feature extraction

The novel framework utilizes the extraction of the emotion tags from the tweets. The emotion extraction module leverages the six emotion tags: joy, sadness, anger, fear, love,

---

**Algorithm 1** Sentiment feature extraction module

*Input: $e_x$ : TextualModalityT$_i$*

*Output: $S_i$: Sentiment Label*

**function** Sentiment($e_x$)

  1:   $S_v = IntensityAnalyzer()$

  2:   $S_d = S_v.polarity\_scores(T_i)$

  3:   $S_i = polarity\_to\_text(S_d)$

  4:   **return** $S_i$

---

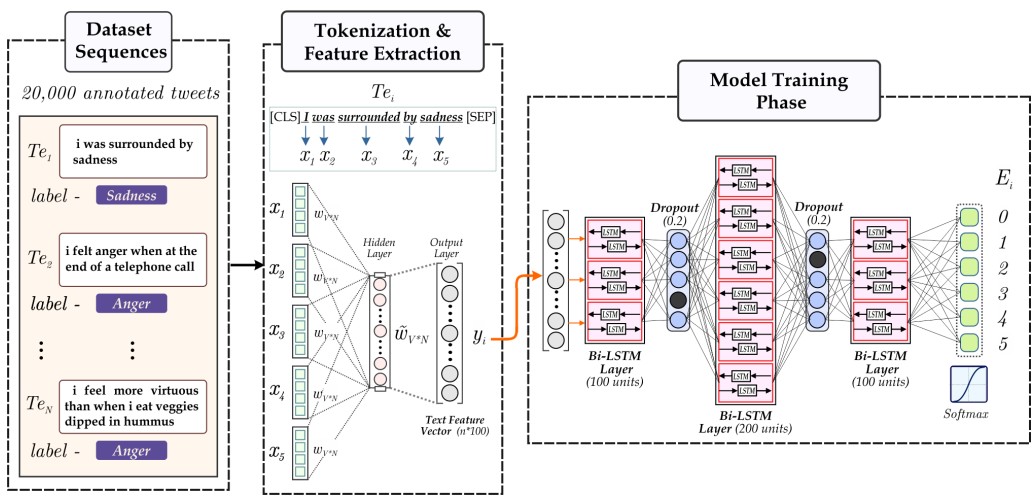

**Figure 5**   **Illustration the RNN based emotion extraction module.**

                    

and surprise. We employ a RNN-based deep learning model, *i.e.,* Bi-directional LSTMs, to associate each tweet with an emotion. Leveraging the 'Emotions dataset for NLP', we classify each tweet into one of six pre-defined emotion categories based on its textual content. Figure 5 accordingly depicts the process employed for emotion extraction from the textual data of tweets.

    To extract emotion tags, we utilize the RNN-based module, namely Bi-directional LSTM, consisting of a cell, input gate, output gate, and forget gate (*Van Houdt, Mosquera & Nápoles, 2020*). Every gate follows a function for the calculation of output values corresponding to it. Equations (5), (6) and (7) represents the required functions. The module has recurrently connected blocks. The Eq. (5) describes the forward pass limiting the input in the RNN network where the current input is, say $x^t$ and $z^{t-1}$ is the output of LSTM in the last iteration.

$$y^{(t)} = \mathcal{F}(W_y x^t + R_y z^{t-1} + b_y) \tag{5}$$

    The $\mathcal{F}$ in Eq. (5) usually is *tanh* whereas $W_y$ and $R_y$ are weights associated with $x^t$ and $z^{t-1}$ respectively, while $b_y$ represents the bias weight vector. The given function allows the

calculation of the forward pass of the LSTM architecture. For the next connected block, the current input is combined with the previous layer's output, as shown in Eq. (6) followed by the removal of information from the previous cell, *i.e.*, $g^{(t)}$, following the same procedure as input gate on current input, previous cell output and the state $c^{t-1}$. The $\tau$ is always *sigmoid* and $p$, $W$, $R$ are weights at respective stages.

$$i^{(t)} = \tau(W_i x^t + R_i z^{t-1} + p_i \odot c^{t-1} + b_i) \tag{6}$$

Equation (6) represents the function of the input gate. By combining the $y^{(t)}$, $i^{(t)}$ and $g^{(t)}$ we can calculate the cell value as $c^{(t)} = y^{(t)} \odot i^{(t)} + c^{(t-1)} \odot g^{(t)}$. This $c^{(t)}$ corresponds to the function of the cell gate. Finally, the output of the recurrent model can be described as in Eq. (7).

$$O^{(t)} = \tau(W_o x^t + R_o z^{t-1} + p_o \odot c^t + b_o) \tag{7}$$

Our emotion extraction module leverages two LSTM networks, one in the forward direction and the other in the backward direction, to capture the contextual insights of the tweet in order to generate the output label. Figure 5 accordingly depicts the process employed for emotion extraction from the textual modality. The process initiates with passing the raw text $T$ (representing individual tweets) as input to the model. $T$, when tokenized, gives $T_1$, which provides us with an embedding vector by utilizing the GloVe model of the gensim library. GloVe stands for Global Vectors for word embedding. It is a pre-trained model trained on extensive text data; utilizing this, we get our embedding vectors for available textual modality (*Pennington, Socher & Manning, 2014*).

Furthermore, the embedding passes through bidirectional recurrent blocks of dimension $100, 200, 100$ as depicted in lines $3, 5, 7$ of the algorithm. Finally, the output of the last bidirectional LSTM layer is passed through a fully connected dense layer of dimension 6 followed by softmax activation function to get label $E_i$ where $E_i \in joy, sadness, surprise, love, anger, fear$.

The algorithm discussed below shows flow the model follows.

---

**Algorithm 2** Emotion extraction module

---

*Input:* $T$: Tweet from Emotion dataset

*Output:* $E_i$ : Emotion label

**function** Emotion($T$)

1:   $T_1 \leftarrow clean\_text(T)$

2:   $T_1 \leftarrow TokenizeEmbed(T)$

3:   $g_1 \leftarrow Bi(LSTM(T_1, 100))$

4:   $g_1 \leftarrow dropout(0.2)$

5:   $g_2 \leftarrow Bi(LSTM(g_1, 200))$

6:   $g_2 \leftarrow dropout(0.2)$

7:   $C \leftarrow Bi(LSTM(g_2, 100))$

8:   $E_i \leftarrow Dense(6, activation = "softmax")$

9:   **return** $E_i$

---

## Classification

The final classification module employs a neural network architecture as illustrated in Fig. 3. This module integrates emotion and sentiment tags with the tweet content obtained from the datasets. The concatenated representation first undergoes a textual feature extraction module, which provides a vector representation of the concatenated textual modality. Furthermore, the vector serves as the input to the classifier, which subsequently generates the desired label.

At this stage, the updated feature vector has text appended with emotion and sentiment tags, which was output for the preceding emotion extraction module and sentiment extraction algorithm. The encoded feature vector is of dimension 768 for each tweet. Furthermore, the vectors are passed through dense layers with ReLU between the layers to add non-linearity. Equation (8) demonstrates the dense layer process.

$$z = Wx + b \tag{8}$$

Equation (8) represents the working of a dense layer, performing a linear transformation on the input data. The weight matrix $W$ depicts the significance of each input element, while the bias vector $b$ introduces activation among neurons. Within a neural network architecture, dense layers leverage a weighted linear combination of their inputs, augmented by a bias term, to generate a new representation of the incoming data, potentially enabling the network to extract more complex features or relationships.

$$P = softmax_j(y_j) \tag{9}$$

Lastly, the features $(y_j)$ are passed through a final dense layer of dimension two, followed by the softmax activation function to get the probabilities of the label of the tweet as demonstrated in Eq. (9).

---

**Algorithm 3** Classification module

---

*Input: T*: Tweet with emotion and sentiment tag
*Output: L*$_i$ : Rumor or Non-rumor label
**function** Classifier($T$)

  1:  $T_1 \leftarrow clean\_text(T)$

  2:  $T_2 \leftarrow TokenizeEmbed(T_1)$

  3:  Define hidden layer activation function: $f(x) = \text{ReLU}(x)$

  4:  **Function**: Forward pass $(x)$

  5:  $\mathbf{z} = \mathbf{W}_1\mathbf{x} + \mathbf{b}_1$

  6:  $\mathbf{h} = f(\mathbf{z})$

  7:  $\mathbf{y} = \mathbf{W}_2\mathbf{h} + \mathbf{b}_2$

  8:  **return y**

  9:  **Function**: Predict class $(T_2)$

10:  $\mathbf{y} = $ Forward pass$(T_2)$

11:  $i = softmax_j(\mathbf{y}_j)$

12:  **return** $L_i = \text{L}(\mathbf{i})$

---

Finally, the label can be procured by calculating the maximum probabilities denoted as $P$.

The algorithm given below demonstrates how the classifier module works. As stated in the algorithm, the classifier utilizes two linear layers with ReLU as an activation function to add non-linearity. The initial linear layer takes 768 as the input dimension and gives an output of dimension 50. After adding non-linearity by ReLU, the other linear layer takes input of 50 dimension from the previous layer and gives away output of dimension 2 corresponding to the two output labels.

# EXPERIMENTATION AND RESULTS

In this section, we will assess the effectiveness and precision of our model by conducting experiments on various features incorporated within it. Additionally, we will demonstrate the model's performance using two datasets, a publicly available dataset, *i.e.,* PHEME, and a real-time dataset, *i.e.,* Twitter24. This section will encompass the necessary setup details, parameter analysis, feature explanations, and a conclusive comparison. Online IDEs are utilized for this work. In our work, we have experimented with only textual data.

## Aggregation of textual data

We have utilized the publicly available "PHEME" dataset and a novel dataset named "Twitter24" in our work. Following subsections depicts the datasets description.

### PHEME

The dataset is based on actual life incidents that happened around the world; the events are defined as hashtags, namely #charliehebdo, the incident of firing in France, and #ferguson, an incident of killing a black person in the USA. The dataset was formed consisting of a total of nine events. The tweets were taken from around 25,691 Twitter (X) users. This work utilizes the events mentioned above.

### Twitter24

The novel dataset named "Twitter24" has been curated from the real-time tweets extracted manually from the social media platform Twitter (X). The dataset consists of only textual modality. It consists of tweets from popular user accounts like "Narendra Modi", "Virat Kohli", focusing more on information circulating in India. The labels are assigned manually and the correctness is established by utilizing fact checking website *i.e.,* "Boom Fact Check". The purpose of this dataset is to validate the performance of SEMTEC model on real-time data.

As referred in Table 2, the "PHEME" dataset consists of a total of 62,445 tweets and "Twitter24" consists around 4,829 tweets which are distributed in two labels, *i.e.,* rumor and non-rumor. Three mutually exclusive training, testing, and validation sets are created from the tweets with tweet share as 70%, 20%, and 10%, respectively.

For training our RNN-based deep learning module, "Emotion dataset for NLP" is utilized.

**Table 2  The table illustrates the count of different parameters that define the datasets.**

| Parameters | PHEME | Twitter24 |
|---|---|---|
| No. of users | 25,691 | 4,200 |
| No.of tweets | 62,445 | 4,829 |
| No. of rumors | 13,824 | 2,782 |
| No. of non-rumors | 48,619 | 2,043 |

**Table 3  Overview of "Emotion dataset for NLP" with textual data and labels.**

| Tweet | Label |
|---|---|
| i was feeling a little vain when i did this one | sadness |
| i felt anger when at the end of a telephone call | anger |

Table 3 illustrates an overview of the dataset. The dataset aggregates a total of 20,000 tweets, categorized into six different classes, namely joy, sadness, anger, fear, love, and surprise, with tweet counts as 6,761, 5,797, 2,709, 2,373, 1,641 and 719 respectively.

## Pre-processing the dataset

This section presents the data pre-processing steps to address inconsistencies within the dataset and reduce the potential for erroneous outcomes in subsequent analyses. The raw data from the dataset consists of redundancy and inconsistencies that must be addressed. Eq. (10) illustrates removing undefined values from the dataset denoted as $D$.

$$D_f = drop\_na(D) \tag{10}$$

Furthermore, duplicate redundancy can be removed using $drop\_duplicates()$.

## Setup requirements for comparitive analysis

This section includes a detailed description of the system and software requirements required to reproduce the results provided in this work. We present the specifications clearly and concisely using tables for easy reference. Leveraging the given parameters, the reproducibility of the mentioned results can be achieved.

### Software requirements

This section details the computational environment that facilitated the research and enabled the achievement of the presented results.

Table 4 illustrates all the software parameters used for setting up the running environment of the proposed SEMTEC method.

### Hardware requirements

The following section details the hardware requirements used in this study to ensure the reproducibility of the presented results. We focus on the critical hardware components that significantly impact the performance of our experiments. Additionally, we acknowledge that similar configurations with comparable capabilities might achieve similar outcomes, aiming to broaden accessibility for researchers with varying resource constraints.

**Table 4  Software specifications for the SEMTEC method.**

| Parameter | Value |
| --- | --- |
| IDE | Kaggle |
| Disk Space | 73.1 GB |
| RAM (CPU) | 30 GB |
| RAM (GPU) | 15 GB |
| GPU Type | Nvidia Tesla T4 |
| No. of Accelarator | 02 |
| Total RAM (with accelerator) | 15 + 15 + 30 |
| CPU | Intel Skylake/AMD/Broadwell |
| No. of CPU Cores | 04 |

**Table 5  Hardware specifications for completing the proposed work.**

| Parameter | Value |
| --- | --- |
| Device | Lenovo IdeaPad L340 |
| Processor | Intel Core i7 |
| Generation | 9th Gen |
| Installed RAM | 8.0 GB |
| Operating system | Windows |
| Edition | Windows 11 |
| Disk space | 1 TB |
| SSD | 256 GB |

Table 5 demonstrates the specifications of the local system utilized in fulfillment of the proposed SEMTEC method.

## Compared methods

To evaluate the effectiveness of our proposed model, we compare its performance to existing methods. This section details the various methods employed in our experimentation. We compare our proposed SEMTEC method with different Deep Learning-based models such as FastText Classifier (*Joulin et al., 2016*), GAN-GRU (*Ma, Gao & Wong, 2019*), TDRD (*Xu, Sheng & Wang, 2020*), BiGCN (*Bian et al., 2020*), GCAN (*Lu & Li, 2020*) and GACL (*Sun et al., 2022*).

### *FastText classifier*

FastText Classifier (*Joulin et al., 2016*) represents text data as a bag of words and employs a linear classifier following training. This approach aligns with establishing simple machine-learning models as strong baselines for text classification tasks.

### *GAN-GRU*

The GAN-GRU (*Ma, Gao & Wong, 2019*) method is based on Generative Adversarial Network(GAN). It employs a generator to introduce conflicting and uncertain perspectives into the original tweet thread, leading the discriminator to learn from more complicated examples.

**Table 6  Illustrating the use of features in different methods for rumor detection task.** The "X" indicates that feature is not utilized while "Y" indicates that corresponding feature is used in the mentioned work. The results for the proposed model are shown in bold.

| Method | Contextual analysis | Sentiment tags | Emotion tags | Propagation feature | Text classifier |
|---|---|---|---|---|---|
| FastText | X | X | X | X | Y |
| GAN-GRU | Y | X | X | X | Y |
| TDRD | Y | X | X | X | Y |
| UDGCN | X | X | X | Y | Y |
| GCAN | Y | X | X | Y | Y |
| BiGCN | X | X | X | Y | Y |
| GACL | Y | X | X | X | Y |
| **SEMTEC** | **Y** | **Y** | **Y** | **X** | **Y** |

### TDRD

The TDRD (Topic Driven Rumor Detection) method extracts the post's topic to derive the tweet's label. *Xu, Sheng & Wang (2020)* first automatically perform topic classification on source microblogs, and then they successfully incorporate the predicted topic vector of the source microblogs into rumor detection.

### BiGCN

BiGCN (Bi-directional Graph Convolutional Network) (*Bian et al., 2020*) method utilizes both propagation and dispersion for rumor detection. The model incorporates both features by operating from bottom-to-top and top-to-bottom propagation of rumors. The up-down GCN(UD-GCN) incorporates the propagation features, whereas bottom-up(BU-GCN) deals with the dispersion of rumor.

### GCAN

The GCAN (Graph Aware Co-attention Networks) (*Lu & Li, 2020*), a neural network-based method, predicts whether the tweet is accurate and simultaneously generates explanations highlighting the evidence from suspicious retweeters and the concerning words they use.

### GACL

GACL (Graph Adversarial Contrastive Learning) (*Sun et al., 2022*) deals with poor generalization in conventional models, where the module of contrastive learning extracts similarities and differences among tweet threads. Furthermore, the AFT(Adversarial Feature Transformation) module generates conflicting samples to extract event-invariant features. Table 6 compares features among the existing research and the proposed SEMTEC method.

Existing rumor detection methods primarily rely on classification approaches, focusing on features extracted from follow-up comments to the initial tweet as indicated in Table 6. However, these methods often neglect the potential value of the primary tweet itself for early rumor detection, particularly in real-time scenarios. This paper introduces SEMTEC, a novel approach that moves beyond classification and emphasizes the importance of the primary tweet. SEMTEC leverages a comprehensive feature set that incorporates

**Table 7 Parameters utilized for comparision.**

| Parameters | Value |
|---|---|
| Learning rate | 5e−5 |
| Epoch | 30 |
| Optimizer | AdamW |
| No. of lables | 2 |

functionalities employed in prior work and introduces additional features to enhance real-time detection accuracy.

## Evaluation metrics

This study evaluates the performance of compared approaches to measure their efficiency. We quantify the efficacy using specific metrics: Accuracy, F1-score, Recall, and Precision.

   We define the precision concerning a particular class where $label \in \{rumor, non-rumor\}$, as the quotient of the number of correctly predicted instances of that label divided by the total number of predictions made for that label. This is mathematically represented in Eq. (11).

$$Precision_{label} = \frac{True\_Predicted_{label}}{Total\_Predicted_{label}} \qquad (11)$$

In the context of classification tasks, recall serves as a crucial metric to assess the sensitivity of a classifier. We precisely measure the effectiveness of the classifier in identifying true positives. Recall quantifies the proportion of actual positive (rumor) instances the classifier correctly classified. We further formalize this in Eq. (12).

$$Recall_{label} = \frac{True\_Predicted_{label}}{Total_{label}} \qquad (12)$$

We leverage the F1-score metric for combining precision and recall into a single, balanced measure. The F1-score is formulated as the harmonic mean of these two metrics in Eq. (13). Through this, we aim to provide a comprehensive evaluation of our classifier's performance, considering its ability to correctly identify positive instances (precision) and avoid false negatives (recall) against the considered existing approaches.

$$F1-score_{label} = \frac{2 \times Precision_{label} \times Recall_{label}}{Precision_{label} + Recall_{label}} \qquad (13)$$

Accuracy is a metric that states the overall performance of the mode. In our work, accuracy can be stated as the average precision calculated for available labels.

## Comparison parameters

This section discusses the parameters utilized for comparing the performance of proposed SEMTEC method with existing research. The parameters are listed in Table 7.

   Table 7 illustrates the required parameters involving learning rate, optimizer, and number of epochs. This will facilitate the comparison of SEMTEC with existing research.

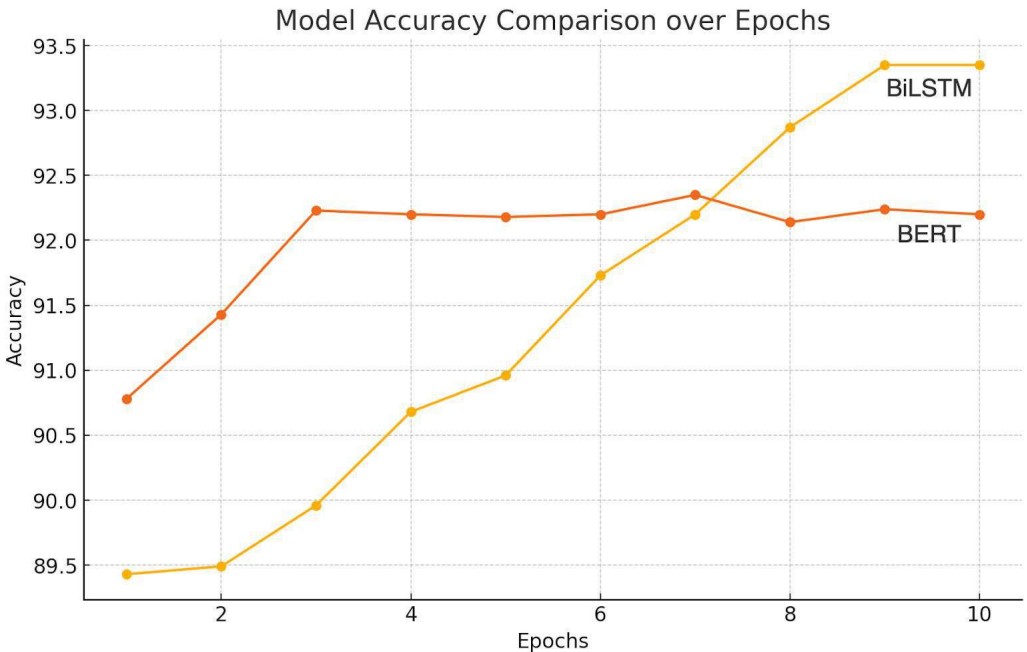

**Figure 6** **Illustration of accuracy comparison for emotion extraction.**

## Results

This section discusses the results of the experiments to evaluate the proposed model. We compare the performance of the proposed model against existing methods to assess its effectiveness. The comparison is further depicted to illustrate the relative performance of each method. The values presented are in the range of 0 to 1, and the parameters calculated have results for every class of categorization, namely Rumor (R) and Non-rumor (N). Before proceeding to the effectiveness comparison of the SEMTEC method, it is necessary to look at the effectiveness of the sub-unit of SEMTEC, *i.e.,* Emotion Extraction Methods.

### Evaluation of emotion extraction methods

To optimize the work and increase the efficiency of SEMTEC, we focused on identifying the best method for emotion extraction from the textual modality. We experimented with RNN-based deep learning methods and encoder-based deep learning methods. Standard BERT is utilized as an encoder model for the comparison. It takes into account the ambiguous meaning of the textual modality that enhances its performance in NLP tasks (*Kenton & Toutanova, 2019*).

Figure 6 shows that the RNN-based methods best identify the emotion from the available textual modality. Bi-LSTMs are acknowledged for their capability to capture contextual information due to their ability to process sequences in both forward and backward directions. This characteristic allows the model to extract emotions while considering the

**Table 8  Effectiveness comparision results from exiting methods on "PHEME" dataset.** The results for the proposed model are shown in bold.

| Model | Precision | | Recall | | F1-score | | Accuracy |
|---|---|---|---|---|---|---|---|
| | **R** | **N** | **R** | **N** | **R** | **N** | |
| FastText | 0.00 | 0.66 | 0.00 | **1.00** | 0.00 | 0.79 | 0.66 |
| GAN-GRU | 0.77 | 0.79 | 0.79 | 0.76 | 0.78 | 0.77 | 0.78 |
| TDRD | 0.81 | 0.83 | 0.63 | 0.92 | 0.71 | 0.87 | 0.82 |
| UDGCN | 0.75 | 0.83 | 0.67 | 0.87 | 0.70 | 0.85 | 0.80 |
| GCAN | 0.76 | 0.87 | 0.75 | 0.87 | 0.76 | 0.87 | 0.83 |
| BiGCN | 0.75 | 0.86 | 0.73 | 0.87 | 0.74 | 0.86 | 0.82 |
| GACL | 0.80 | 0.87 | 0.75 | 0.90 | 0.77 | 0.88 | 0.85 |
| SEMTEC | **0.91** | **0.92** | **0.94** | 0.90 | **0.93** | **0.91** | **0.92** |

surrounding context within each tweet. This led us to move forward with Bidirectional LSTM (RNN-based method) for emotion extraction.

The experiment compared the stated emotion extraction methods. A standard optimization algorithm, Adam, was employed alongside categorical cross-entropy loss and a softmax activation function. The training process was executed for approximately 35 epochs.

### Effectiveness comparisions

To evaluate the efficacy of our proposed approach for rumor detection, we compare its performance to existing techniques. We evaluate the performance of various techniques based on established metrics like Precision, Recall, F1-score, and Accuracy. Leveraging the publicly available "PHEME" dataset and the real-time dataset "Twitter24", we illustrate the effectiveness of our model by comparing its results to those obtained using previously employed techniques.

Table 8 shows how significantly better our SEMTEC model performs on the datasets, as mentioned earlier, than the prior techniques. The performance of our SEMTEC model on different metric parameters, namely Precision, Recall, and F1-score, is 0.91, 0.92, and 0.92, respectively, on the PHEME dataset. In terms of accuracy, we achieve a surge of around 0.7 from the best existing method.

Figure 7 illustrates a comprehensive visualization of the variations in F1-score across different models. Our model achieves superior performance on PHEME due to its incorporation of emotion and sentiment features alongside a contextual analysis of textual modalities, as opposed to current techniques, which rely on the textual content of social media posts. Figure 8 illustrates the qualitative assessment of the proposed work. The SEMTEC surpasses the existing research, and this can be justified by the feature set and deep learning models utilized in this work.

Furthermore, we have compared our work with the existing classifier. The classifiers can be divided into machine learning-based classifiers like support vector machine (SVM), Random Forest(RF) and deep learning-based classifiers like Transformer and Bi-directional LSTMs.

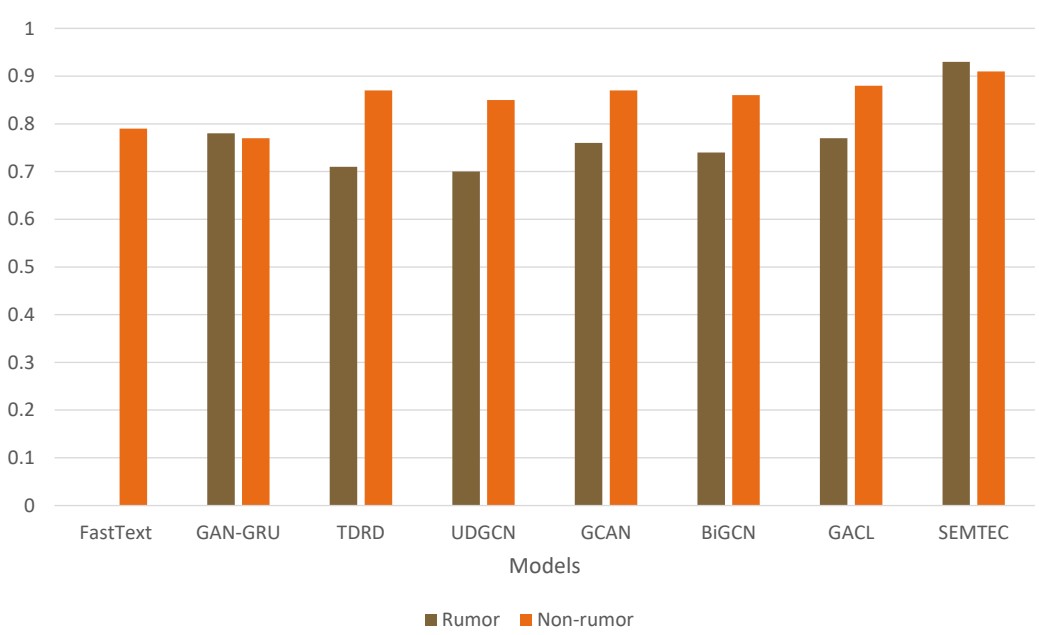

**Figure 7** Illustration of accuracy comparison for diverse models on the PHEME dataset.

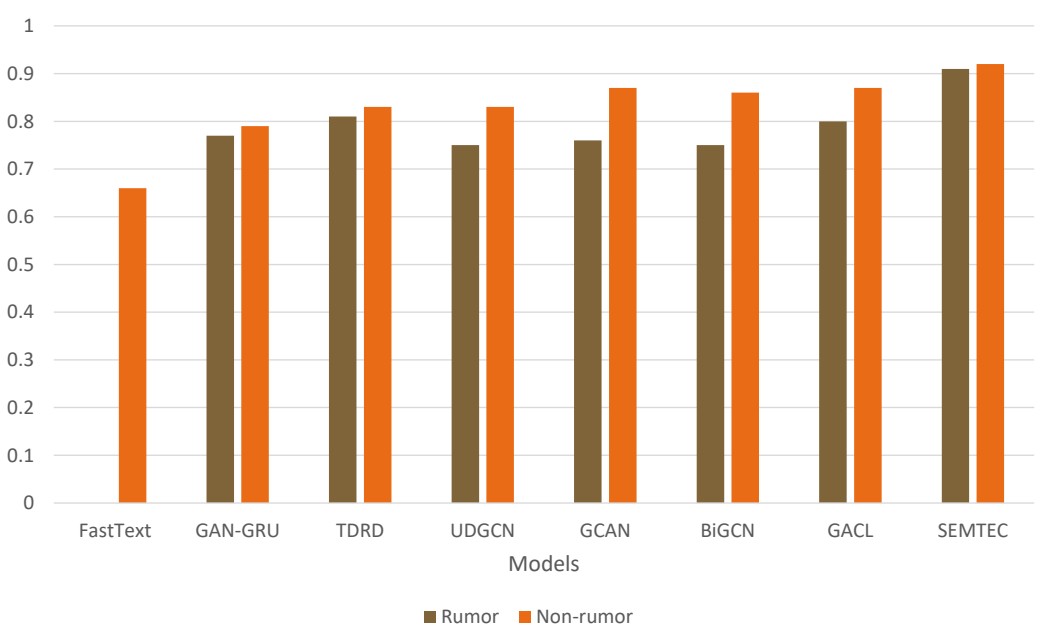

**Figure 8** Illustration of precision comparison for diverse models on the PHEME dataset.

**Table 9 Effectiveness comparision of SEMTEC on PHEME dataset with standard classifiers.** The results for the proposed model are shown in bold.

| Model | Precision | | Recall | | F1-score | | Accuracy |
|---|---|---|---|---|---|---|---|
| | R | N | R | N | R | N | |
| Naive Bayes (NB) | 0.72 | 0.75 | 0.66 | 0.80 | 0.68 | 0.77 | 0.74 |
| Random Forest (RF) | 0.68 | 0.74 | 0.65 | 0.77 | 0.67 | 0.75 | 0.72 |
| Support Vector (SVM) | 0.74 | 0.76 | 0.66 | 0.82 | 0.69 | 0.79 | 0.75 |
| BiLSTM | 0.68 | 0.75 | 0.67 | 0.76 | 0.66 | 0.75 | 0.72 |
| Transformer | 0.67 | 0.74 | 0.66 | 0.76 | 0.67 | 0.73 | 0.71 |
| SEMTEC | **0.91** | **0.92** | **0.94** | **0.90** | **0.93** | **0.91** | **0.92** |

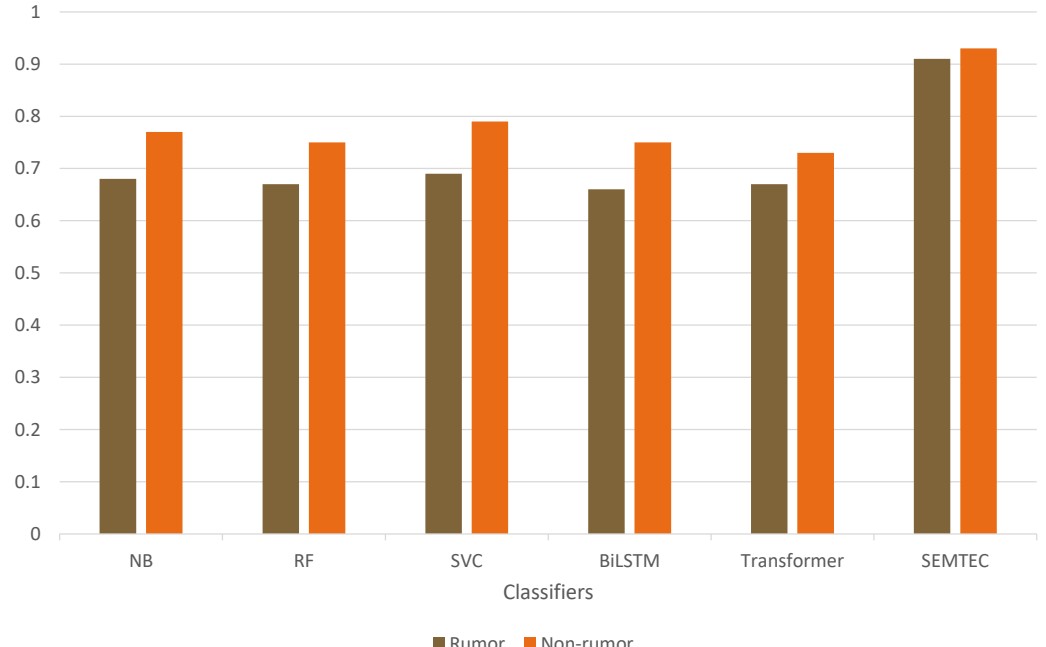

**Figure 9 Illustration of accuracy comparison for standard classifiers on the PHEME dataset.**

As illustrated in Table 9, the SEMTEC method outperforms the standard classifiers with a significant difference. The findings justify the relationship between the semantic attributes and the veracity of the tweet, which aids in the classification task. Figure 9 visually illustrates the performances of standard classifiers on "PHEME" dataset. Our findings suggest that rumor detection extends beyond a simple classification task.

To validate the performance of our proposed SEMTEC method, we further experimented with the novel "Twitter24" dataset. The experimentation demonstrates that the proposed SEMTEC method surpasses the existing standard methods used for classification by around 2%. Table 10 illustrates the findings highlighting the superior performances.

**Table 10  Table representing performance of SEMTEC on real time Twitter24 dataset.** The best results are shown in bold.

| Model | Precision | | Recall | | F1-score | | Accuracy |
|---|---|---|---|---|---|---|---|
| | R | N | R | N | R | N | |
| Naive Bayes (NB) | 0.90 | 0.84 | 0.87 | 0.87 | 0.89 | 0.85 | 0.87 |
| Random Forest (RF) | 0.88 | 0.87 | 0.90 | 0.84 | 0.89 | 0.86 | 0.88 |
| Support Vector (SVC) | 0.90 | 0.92 | 0.95 | 0.86 | 0.92 | 0.89 | 0.91 |
| BiLSTM | 0.86 | 0.91 | 0.94 | 0.80 | 0.90 | 0.85 | 0.88 |
| Transformer | 0.88 | 0.84 | 0.80 | 0.85 | 0.88 | 0.85 | 0.87 |
| SEMTEC | **0.92** | 0.92 | 0.95 | **0.89** | **0.94** | **0.91** | **0.93** |

### Performance gain analysis

In this section, we analyze the effectiveness of our SEMTEC method with a combination of various features utilized in our work. The performance of our model is the outcome of integrating emotion features, sentiment features, and contextual analysis of text. *Ajao, Bhowmik & Zargari (2019)* also showed the interconnectedness of semantic features with detecting fake news. We conducted more experiments to justify that the proposed SEMTEC method establishes the relationship between semantic attributes and the veracity of the tweet. To justify the findings and the relationship between the semantic features, *i.e.,* sentiment and emotion features, we conducted an ablation study demonstrating semantic features effect on the identification of rumor.

Figure 10 demonstrates our method's performance by including various features. We discuss our proposed model's performance by including emotion and sentiment tags along with the textual modality. The emotion feature directly conveys the tweet's objective. This variant is mentioned as SEMTEC. This model performs better than the prior SEMTEC - (E), where only text and sentiment tags were used. SEMTEC - (E+S) illustrates the textual modality without any features. This work presents the results achieved in terms of accuracy. This enhancement can be attributed to the incorporation of emotion and sentiment tags as they facilitate a deeper understanding of the sentence semantics, which further prove significant in predicting the sentiment polarity of the post.

### Analysis of curated dataset with emotion labels

In this section, we discuss the curated "EmoPHEME" dataset. Our proposed SEMTEC method utilizes emotion tags extracted using RNN based deep learning model, *i.e.,* Bidirectional LSTM. This module, trained on the "Emotion dataset for NLP", enables the generation of emotion labels for the "PHEME" dataset, capturing the emotional aspect of the tweets. The emotion extraction module facilitates exploring new informational dimensions within the transformed "PHEME" dataset named "EmoPHEME". This enhanced dataset offers potential applications in training and testing models designed for emotion-related sentiment analysis tasks. Figure 11 visualizes the distribution of the emotion labels across the new "EmoPHEME" dataset. The labels, namely joy, anger,

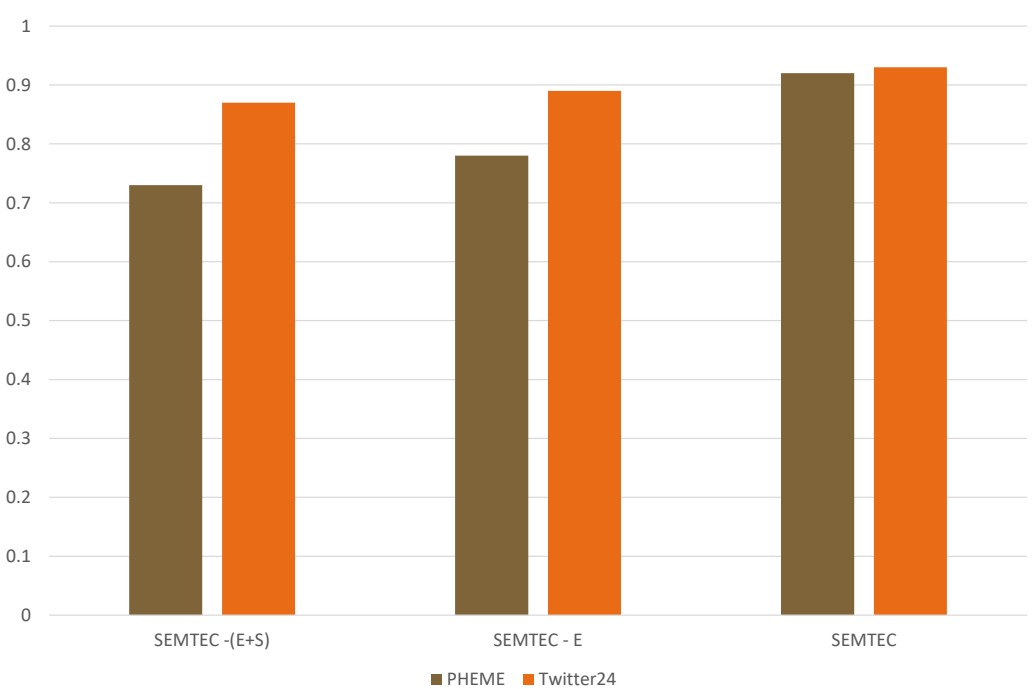

**Figure 10** Illustration of performance of SEMTEC for various variants on the PHEME and Twitter24 datasets.

sadness, love, surprise, and fear, have percentage division as 29%, 20%, 27%, 5%, 1% and 18%, of the total tweets, respectively.

## DISCUSSION

This paper proposes a sentiment and emotion-driven transformer classifier approach for rumor detection on Twitter (X) posts using deep learning methods. This work introduces an intrinsic model that estimates the sentiment and emotion of any meaningful sentence in the English language. Our proposed method automatically evaluates sentiment and emotion within short, independent text segments, such as tweets. Given the rise of social media posts incorporating modalities beyond text, this study investigates the combined role of textual content, sentiment, and emotion in accurately assessing tweet veracity. This study prioritizes the significance of the primary tweet in veracity assessment, as the keywords within the text play a significant role in enabling the model to judge the conveyed emotion and sentiment contained in the text, leading to the verdict on the tweet's truthfulness.

This study proposes a method that leverages the inherent relationships within the textual data. We use the concatenation ahead to include the semantic aspect in the main tweet.

The proposed SEMTEC method outperforms the existing models. Additionally, the experimental outcomes on various baseline classifiers like Random Forest, Bi-LSTM, and Transformer-Based Models demonstrate the effectiveness of our model for text-based tweets.

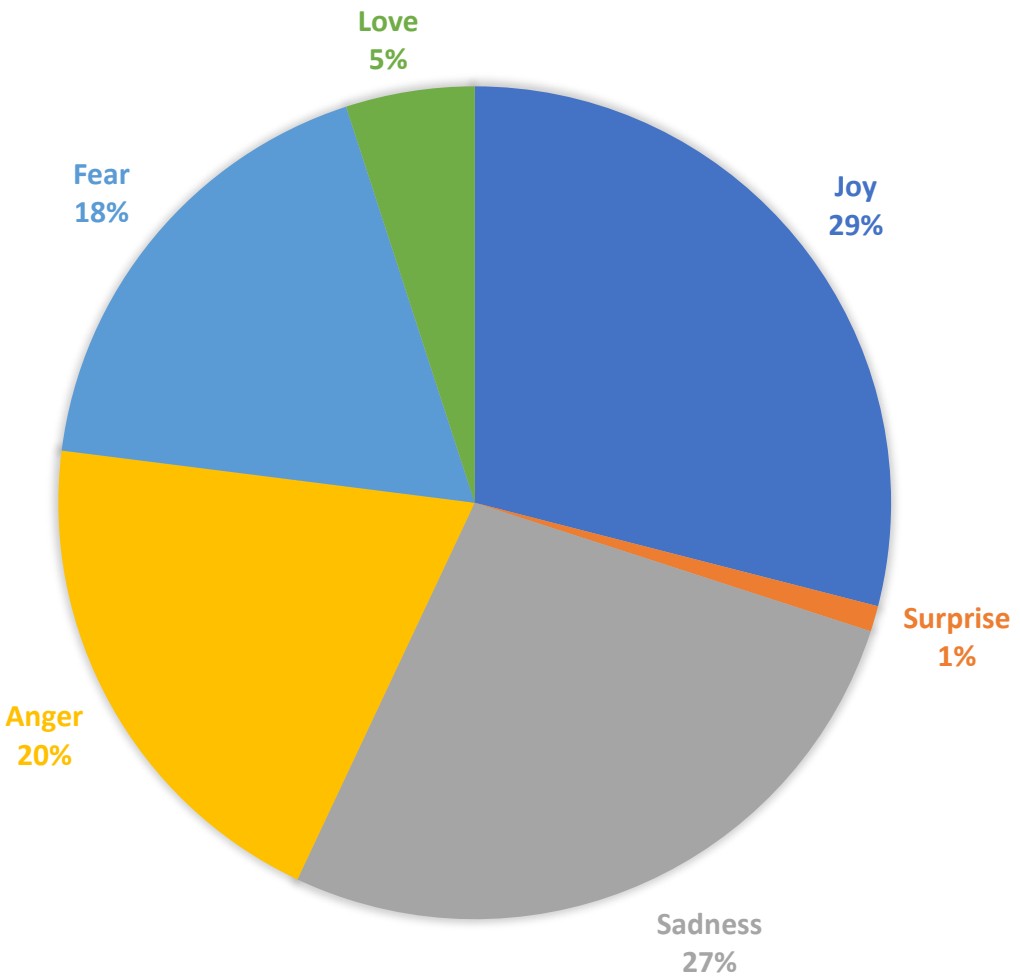

**Figure 11** Distribution of emotion lables on the curated "EmoPHEME" dataset.

## Pros and cons of SEMTEC method

This section illustrates some pros following the practical implementation of the SEMTEC method. We tried to be inclusive and unique with our work, but some aspects might get little attention; those will also be mentioned in this section.

### Pros and practical implementation of SEMTEC method

- The SEMTEC method reveals the contextual relationship hidden inside the root or main tweet. It considers the emotional aspect of the tweet, aiding in correctly providing a label to the tweet.
- The proposed method can be used in various social media platforms and publication industries for rumor identification and detection.

***Cons or under considered aspect of SEMTEC***
- In previous literature, the authors utilized follow-up comments as a propagation feature. Our proposed work mainly focused on the root or main tweet. The propagation feature can be utilized as an additional feature in the extension of this work.
- The real-time streaming of data involves processing data as soon as it gets pushed on any platform. The model pipeline should not hold any lagging(time delay) when processing this kind of data. Our SEMTEC method involves numerous modules, such as preprocessing and feature extraction units, before providing a label to the tweet. This might cause a delay in processing data in real time.

## FUTURE WORK

The aspects of future extension of our proposed work are illustrated below in subsections:

### Semantic web approach for rumor detection

The proposed SEMTEC method for detecting rumors can be extended to utilize the semantic web technology to identify rumors in real time. Semantic web is an ontology-based technique that uses queries to solve problems.

The "semantic" in the semantic web stands for machine processable or how machines can utilize data, whereas the "web" depicts interconnected objects mapped *via* URIs to the resources. In simple terms, the semantic web is an idea related to the extension of WWW, *i.e.,* the World Wide Web, which can provide software with metadata of the information and published data. We plan to utilize the knowledge graph representation of platforms like Wikidata and DBpedia *via* semantic web in the extension of our proposed work. Through the SPARQL queries, articles can be accessed in real-time and help validate whether the tweet or information is a rumor.

### Rumor detection on low resource language

The extension of the proposed work takes into consideration the low-resource languages. The future work addresses the challenge of rumor detection in low-resource languages, explicitly focusing on Hindi, a widely spoken language in India despite the nation's multilingual landscape. This research undertakes the development of a Hindi rumor detection model. We built a dataset leveraging Hindi tweets extracted from social media platform X (previously known as Twitter). The dataset was annotated using established fact-checking websites. Furthermore, we are trying to build models leveraging emojis and essential features for rumor detection.

## CONCLUSION

This paper presents a novel deep-learning approach for rumor detection in social media microblogs. Our method, the Sentiment and Emotion driven Transformer Classifier (SEMTEC), leverages tweet-level emotions and sentiments for rumor classification.

We employ an RNN-based Bi-directional LSTM model to extract the emotion features and a pre-trained Textblob library to extract sentiment from the tweets. Next, we concatenate the emotion and sentiment tags with the primary tweet. Before subsequent

analysis, the textual data extracted from the dataset is pre-processed, followed by an encoder module to extract contextual features from the pre-processed text. Combined with sentiment and emotional features, these features are fed into a deep-learning model for rumor classification. The proposed method's effectiveness is evaluated on real-world social media datasets comprising English tweets obtained from Twitter. Experimental results demonstrate that our approach outperforms existing methods in terms of performance. The proposed methodology, for now, needs to analyze the propagation nature of the rumor but opens up an opportunity for further exploration.

Furthermore, we can harness the power of the semantic web and knowledge graphs from platforms like Wikidata and DBpedia. This ontology-based approach enables us to access articles in real time that can validate whether a tweet or article is a rumor. Furthermore, we wish to extend our work to a language primarily used in India only, *i.e.,* Hindi.

### Funding
This work was supported by DST, MATRICS Scheme with grant number MTR/2020/000459. The funders had no role in study design, data collection and analysis, decision to publish, or preparation of the manuscript.

### Grant Disclosures
The following grant information was disclosed by the authors:
DST, MATRICS Scheme: MTR/2020/000459.

### Competing Interests
The authors declare there are no competing interests.

### Author Contributions
- Drishti Sharma conceived and designed the experiments, performed the experiments, analyzed the data, performed the computation work, prepared figures and/or tables, authored or reviewed drafts of the article, and approved the final draft.
- Abhishek Srivastava conceived and designed the experiments, analyzed the data, prepared figures and/or tables, authored or reviewed drafts of the article, and approved the final draft.

### Data Availability
  The dataset is available at Kaggle: https://www.kaggle.com/datasets/nicolemichelle/pheme-dataset-for-rumour-detection.
  The raw data and code are available in the Supplemental Files.

### Supplemental Information
Supplemental information for this article can be found online at http://dx.doi.org/10.7717/peerj-cs.2202#supplemental-information.

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
