# Peer review of "Detecting rumors in social media using emotion based deep learning approach"

_PeerJ Computer Science, doi:10.7717/peerj-cs.2202_

## Round 0.1 · original submission · Major Revisions

· Academic Editor

Major Revisions

Dear authors,

Thank you for submitting your article. Feedback from the reviewers is now available. It is not recommended that your article be published in its current format. However, we strongly recommend that you address the issues raised by the reviewers, especially those related to readability, experimental design and validity, and resubmit your paper after making the necessary changes. When submitting the revised version of your article, it will be better to address the following:

1. Please expand the future research directions.
2. The values for the parameters of the algorithms selected for comparison should be given.
3. The paper lacks the running environment, including software and hardware. The analysis and configurations of experiments should be presented in detail for reproducibility. It is convenient for other researchers to redo your experiments and this makes your work easy acceptance. A table with parameter settings for experimental results and analysis should be included in order to clearly describe them.
4. The authors should clarify the pros and cons of the methods. What are the limitation(s) methodology(ies) adopted in this work? Please indicate practical advantages, and discuss research limitations.
5. The research gaps and contributions should be clearly summarized in the introduction section. Please evaluate how your study is different from others in the related work section.
6. English grammar and writing style errors should be corrected. Blank character should be correctly used.
7. Explanation of the equations should be checked. All variables should be written in italic as in the equations. Definitions of variables and the boundaries should be written.

Best wishes,

Reviewer 1 ·

Basic reporting

1. Language needs to be checked through out the paper.
2. The text in the images is not clear.

Experimental design

The paper proposes rumor detection sentiment analysis model using deep learning techniques and there are some gaps in the article that needs to be addressed to get more clarity for the readers.

1. Authors presented SEMTEC model, that works on novel/customized dataset. What about other datasets ? How much preprocessing work is needed for other datasets to adopt this model?

2. Most social media applications are based on the streaming data. How the proposed model works on real time datastreams?

3. Authors used RNN based multi layer model and the details of the RNN model are not given in detail. More details are needed.

4. Provide the detailed table on all the existing works and explain how the proposed model is novel compare to other models

5. How section 4.3.1 is related to proposed approach ? Explanation with example will give more clarity to readers.

6. How the example given in section 5 is processed using the algorithms proposed in section 4. More technical details are needed

7.Is the accuracy of the existing works given in table 4 are based on the “PHEME” dataset ? Then how the novel dataset proposed in section 1 is customized for all the existing works? There is some confusion here and authors needs to justify. Is the proposed work mainly rely on the novel dataset or deep learning models?

8. There are some sentiment analysis models that are energy efficient than RNN based sentiment models which the authors could reference

Validity of the findings

Authors proposed a sentiment analysis model for rumor detection but novelty of the work is missing. There are efficient models to do the same task in the literature and proper justification is needed on why the proposed model can be chosen. Only accuracy is not sufficient to conclude that proposed model works better, other performance metrics also needed to be assed.

Cite this review as

·

Basic reporting

The dataset that was used for this research is a plain text database. The experiment did not properly mention the limitations. I would like to see how the results of the new method called SEMTEC produce results with emojis and conversation order. If we don't take the order of the conversation into account, there is a possibility that the results tag is wrong, which can further be treated as biased results. The way we handle biases is not mentioned in the paper. I want to address that and check the results using the SEMTEC method.
While the authors discussed labels like sad and fear ( sentiments), there is no clear understanding of how the feelings are mapped to the rumors. Rumors are different, and sentiments are different. Rumors spreading appear more attractive and eye-catching to get more visibility for the given tweet. There is a lack of focus on this area.

Experimental design

No comment

Validity of the findings

No comment

Additional comments

No Comment

---

## Round 0.2 · Major Revisions

· Academic Editor

Major Revisions

Dear authors,

Thank you for your submission. We encourage you to address the concerns and criticisms of the reviewers and resubmit your article once you have updated it accordingly.

Best wishes,

Reviewer 1 ·

Basic reporting

Through grammar check is recommended, provide tabular representation between the existing methods and proposed method in section 2. Authors needs to justify why proposed method needs is efficient for sentiment analysis
2. Why authors use RNNs? There 3rd generation neural networks called Spiking neural networks and suggest authors to look at spiking neural networks and if there are any NLP/Sentiment analysis models exists, how it can be integrated to existing work
3. Recommended authors to add some details about datasets and the label size 2 is very small to come to a conclusion that the proposed work is efficient.

Experimental design

Mentioned above

Validity of the findings

No data is provided and the datasets is not efficient and sufficient for experimental analysis

Cite this review as

·

Basic reporting

It is an old problem, and there is no new approach that was discussed in this paper.

Experimental design

Experimental is way too outdated.

Validity of the findings

No satisfied with the findings and results.

Additional comments

NA

---

## Round 0.3 · accepted · Accept

· Academic Editor

Accept

Thank you for clearly addressing all the reviewers' comments. I confirm that the quality of your paper has been improved. The paper now appears to be ready for publication in light of this revision.

Best wishes,

Reviewer 1 ·

Basic reporting

Good

Experimental design

Good

Validity of the findings

Good

Cite this review as